# Cathepsin Z is a conserved susceptibility factor underlying tuberculosis severity

**Rachel K. Meade**[1,2☯], **Oyindamola O. Adefisayo**[1☯], **Marco T. P. Gontijo**[1☯],
**Summer J. Harris**[1], **Charlie J. Pyle**[1], **Kaley M. Wilburn**[1], **Alwyn M. V. Ecker**[1],
**Erika J. Hughes**[1,2], **Paloma D. Garcia**[1], **Joshua Ivie**[3], **Michael L. McHenry**[4],
**Penelope H. Benchek**[4], **Harriet Mayanja-Kizza**[5], **Jadee L. Neff**[6], **Dennis C. Ko**[1,2],
**Jason E. Stout**[7], **Catherine M. Stein**[4,8], **Thomas R. Hawn**[3], **David M. Tobin**[1,2,9],
**Clare M. Smith**[1,2]*

1 Department of Molecular Genetics and Microbiology, Duke University, Durham, North Carolina, United States of America, 2 University Program in Genetics and Genomics, Duke University, Durham, North Carolina, United States of America, 3 Department of Medicine, University of Washington, Seattle, Washington, United States of America, 4 Department of Population and Quantitative Health Sciences, Case Western Reserve University, Cleveland, Ohio, United States of America, 5 Uganda-CWRU Research Collaboration and Department of Medicine, School of Medicine, Makerere University, Kampala, Uganda, 6 Department of Pathology, Duke University, Durham, North Carolina, United States of America, 7 Division of Infectious Disease and International Health, Department of Medicine, Duke University Medical Center, Durham, North Carolina, United States of America, 8 Department of Medicine, Case Western Reserve University, Cleveland, Ohio, United States of America, 9 Department of Integrative Immunobiology, Duke University, Durham, North Carolina, United States of America

☯ These authors contributed equally to this work.
* clare.m.smith@duke.edu

## Abstract

Tuberculosis (TB) outcomes vary widely, from asymptomatic infection to mortality, yet most animal models do not recapitulate human phenotypic and genotypic variation. The genetically diverse Collaborative Cross mouse panel models distinct facets of TB disease that occur in humans and allows identification of genomic loci underlying clinical outcomes. We previously mapped a TB susceptibility locus on mouse chromosome 2. Here, we identify cathepsin Z (*Ctsz*) as a lead candidate underlying this TB susceptibility and show that *Ctsz* ablation leads to increased bacterial burden, pulmonary inflammation and decreased survival in mice. *Ctsz* disturbance within murine macrophages enhances production of chemokine (C-X-C motif) ligand 1 (CXCL1), a known biomarker of TB severity. From a Ugandan household contact study, we identify significant associations between *CTSZ* variants and TB disease severity. Finally, we examine patient-derived TB granulomas and report CTSZ localization within granuloma-associated macrophages, placing human CTSZ at the host–pathogen interface. These findings implicate a conserved CTSZ-CXCL1 axis in humans and genetically diverse mice that mediates TB disease severity.

**Data availability statement:** Summary data are included within the manuscript and supplemental files. Because of the Institutional Review Board (IRB) restriction on the data from Uganda, individual-level data are only available upon request from the Uganda Genetics of TB Data Access Committee (DAC). To initiate a request, contact Dr. Moses Joloba (mlj10@case.edu). For re-analyses of previously published data, relevant repository accession numbers and links are provided in the S1 Data sheet External Data Index.

**Funding:** This work was funded by an NIH Director's New Innovator Award AI183152 (C.M. Smith), a Pew Scholars award (C.M. Smith), and the following NIH grants: AI166304 (D.M.T.), AI127715 (D.M.T. and C.M. Smith), AI181898 (T.R.H. and C.M. Smith), AI162583 (T.R.H., C.M. Stein, and H.M.-K.), N01-AI95383 (C.M. Stein), and T32HL007567 (M.L.M.). M.T.P.G. was supported by a grant (88881.625374/2021-01) from the Fulbright Association and the Coordenação de Aperfeiçoamento de Pessoal de Nível Superior (CAPES). Research reported in this publication was supported in part by the Duke University Center for AIDS Research (CFAR), an NIH funded program (5P30 AI064518). The Duke University BRPC is supported in part by the NIH (P30CA014236). Biocontainment work performed in the Duke Regional Biocontainment Laboratory received partial support for construction and renovation from NIAID (UC6-AI058607 and G20-AI167200) and facility support from the NIH (UC7-AI180254). The sponsors or funders did not play any role in the study design, data collection and analysis, decision to publish, or preparation of the manuscript.

**Competing interests:** The authors have declared that no competing interests exist.

**Abbreviations:** AF, Alexa Fluor; BCG, Bacillus Calmette-Guérin; BF, brightfield; BMDMs, bone marrow-derived macrophages; BP, base pair; BRPC, BioRepository and Precision Pathology Center; CFAR, Center for AIDS Research; Chr, chromosome; Ctsz, cathepsin Z; DAC, Data Access Committee; DO, Diversity Outbred; GEO, Gene Expression Omnibus; GWAS, genome-wide association studies; IACUC, Institutional Animal Care and Use

## Introduction

*Mycobacterium tuberculosis* (*Mtb*), the causative agent of tuberculosis (TB), is a prolific obligate pathogen that has threatened human health for millennia [1]. Through centuries of coevolution, human hosts have developed a plethora of immunological mechanisms in response to *Mtb* infection [2]. Such host-bacterial interactions give rise to a spectrum of disease states, ranging from subclinical infection to fulminant disease [3]. The disease severity experienced by an individual is intricately connected to their genetic background. For example, monozygotic twins are at a demonstrably higher risk for TB concordance than dizygotic twins, highlighting shared genetic identity as a contributor to TB disease outcomes [4–6]. Human genome-wide association studies (GWAS) conducted in impacted geographic regions have also identified polymorphisms that modulate host TB immunity [7–13], indicating numerous immunological pathways involved in *Mtb* susceptibility.

One such gene is cathepsin Z (*CTSZ*), which has been associated with TB susceptibility in independent human studies conducted across Africa. *CTSZ* encodes a lysosomal cysteine protease with a known structure and several reported cellular functions [14–22]. The link between single-nucleotide polymorphisms (SNPs) in *CTSZ* and human TB susceptibility was first established by sibling pair analysis in South African and Malawian populations and independent case-control studies in West Africa [23]. These findings were further validated in a South African case-control study [24] and in a Ugandan GWAS [25] and subsequent household contact study [26]. *CTSZ* is primarily expressed by monocytes and macrophages [27–30] and participates in central immune functions, including dendritic cell maturation [31] and lymphocyte propagation and migration [32,33]. Although in vitro work has been undertaken to study the role of CTSZ in macrophage-driven protection against mycobacteria [34,35], *CTSZ*-linked TB susceptibility has not been explored in vivo. The functional role of CTSZ during *Mtb* infection remains unknown, despite growing genetic evidence of its association with TB disease outcomes.

Studying the mechanisms that underlie *CTSZ*-linked susceptibility in humans is complex [36]. Humans are outbred, and genetic studies of human cohorts must navigate the inherent challenges of natural genetic variation. Moreover, the low- and middle-income countries that harbor 80% of the global TB burden face challenges in and outside of the healthcare sector that complicate TB diagnosis, research, and treatment [37]. The connection between TB severity and host background is not uniquely human. In classic *Mtb* studies measuring postinfection survival, inbred mice have repeatedly illustrated the heritability of TB susceptibility [38,39]. Combining reproducibility with a limited range of genetic variation, classical inbred laboratory mice have served as tractable models that demonstrate the vital impact of host genetic background on *Mtb* pathogenesis. However, because inbred mice are nearly genetically identical within strain [40], studies leveraging standard inbred strains omit the contributions of natural host genetic diversity to TB pathogenesis. Recombinant inbred panels like the biparental BXD [41–44] and octoparental Collaborative Cross (CC) [45–47] systematically model host genetic variation, allowing insight into

Committee; IgG, immunoglobulin; GIRB, Institutional Review Board; IRES, independent ribosomal entry sequence; LD, linkage disequilibrium; LFA-1, lymphocyte function-associated antigen-1; Mac-1, macrophage-1 antigen; MAF, minor allele frequency; Mm, Mycobacterium marinum; MOI, multiplicity of infection; Mtb, Mycobacterium tuberculosis; OADC, oleic acid-albumin-dextrose catalase; PBS, phosphate-buffered saline; PDIM, phthiocerol dimycocerosate; PVDF, polyvinylidene fluoride; QTL, quantitative trait locus; scRNA-Seq, single-cell RNA sequencing; SGCF, Systems Genetics Core Facility; SNPs, single-nucleotide polymorphisms; sPLS-DA, sparse partial least squares discriminant analysis; TB, tuberculosis; Tip5, Tuberculosis ImmunoPhenotype 5; UNC, University of North Carolina.

a spectrum of immune profiles without compromising the reproducibility of inbred strains [48]. We previously reported *Mtb* infection screens of BXD [49] and CC [50] recombinant inbred strains, leveraging these diverse mammalian panels to expand the range of known TB disease complexes and host-pathogen interactions modeled by mice. Using a quantitative trait locus (QTL) mapping approach across a cohort of 52 CC genotypes, we identified a QTL on chromosome 2 (174.29–178.25Mb) significantly associated with *Mtb* burden. Genetic inheritance from NOD/ShiLtJ (NOD), a CC panel founder, at the *Tuberculosis ImmunoPhenotype 5* (*Tip5*) QTL predicted elevated bacterial burden. CC strains that inherited the susceptible *Tip5* variant (*Tip5$^S$*) from NOD succumbed to severe TB prior to the study endpoint. We therefore sought to determine which genes found within the *Tip5* interval could contribute to *Mtb* susceptibility in *Tip5$^S$* CC strains.

Here, we show that CC strains harboring the *Tip5$^S$* locus produce lower levels of CTSZ protein while exhibiting higher bacterial burden than B6 mice following aerosol infection, validating *Tip5* as a susceptibility locus from the large-scale CC cohort screen. We report the first in vivo *Mtb* infections of mice lacking *Ctsz* (*Ctsz$^{-/-}$*). We find that *Ctsz* ablation on a B6 background results in increased *Mtb* burden and an increased risk of mortality following infection. Moreover, *Ctsz$^{-/-}$* mice overproduce CXCL1, a biomarker of active TB [51], at both acute and chronic timepoints. In *Ctsz$^{-/-}$* bone marrow-derived macrophages (BMDMs), we find that CXCL1 is rapidly induced following mycobacterial infection. Leveraging published transcriptional data from genetically diverse mice, humans, macaques, and zebrafish, we find cathepsin Z expression is highest in macrophages following infection. We combine these findings with recent data from a Ugandan patient cohort, highlighting five variants in *CTSZ* as correlates of TB severity. Finally, we identify the presence of CTSZ in CD68[+] macrophages within patient-derived pulmonary granulomas, revealing that CTSZ is produced at the host-pathogen interface in human lungs. Collectively, this work establishes genetic variation in cathepsin Z as a determinant of TB disease outcomes and places human CTSZ in a vital position within the pulmonary microenvironment to impact TB outcomes.

## Results

### Comparative transcriptional analysis to prioritize candidate genes within the *Tip5* locus

We previously reported the *Tip5* QTL (Chr2, 174.29–178.25Mb) as a TB susceptibility locus across the genetically diverse CC panel [50]. To identify gene candidates within *Tip5*, we leveraged published transcriptomic data from *Mtb*-infected mammalian lungs [52,53] (Fig 1A). Within the *Tip5* interval, cathepsin Z (*Ctsz*; alternative names: cathepsin X, cathepsin P) and zinc finger protein 831 (*Zfp831*) were significantly induced in the lungs of genetically heterogeneous Diversity Outbred (DO) mice exhibiting progressive TB, characterized by elevated pulmonary *Mtb* burden and inflammation [52]. In rhesus macaques, animals with progressive TB disease produced significantly more *CTSZ* and *ZNF831* (a high-confidence ortholog of *Zfp831*) transcript in their lungs [52]. In the blood of patients with active TB, *CTSZ* transcription was significantly elevated while *ZNF831*

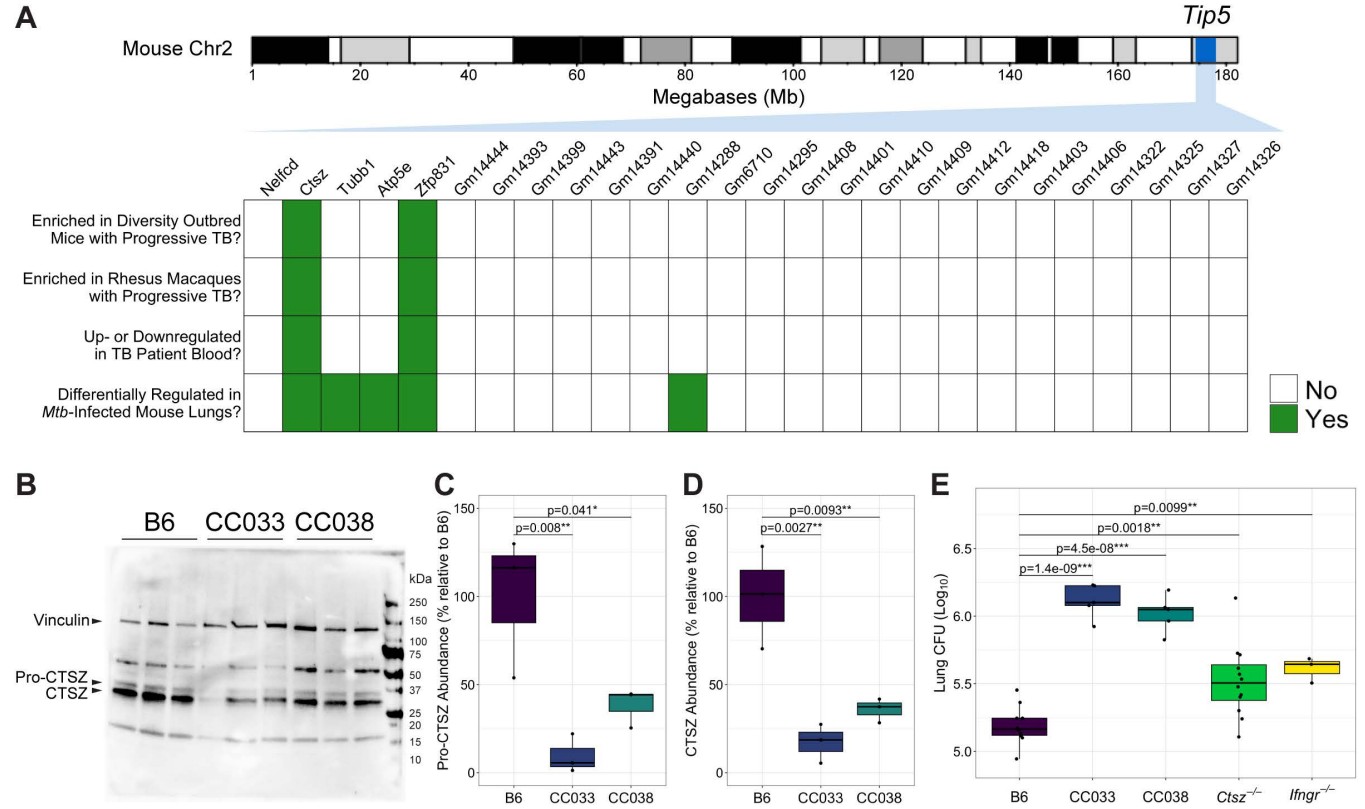

**Fig 1. Identification and validation of *Ctsz* as the lead candidate gene underlying *Tip5*. (A)** Heatmap representation of the per-gene outcome of four distinct criteria for genes within the *Tip5* QTL (95% CI: 174.29–178.25Mb): (i) whether the gene transcript is significantly enriched in the lungs of genetically heterogeneous Diversity Outbred (DO) mice experiencing elevated burden and inflammation after *Mtb* infection [52], (ii) whether the gene transcript is significantly enriched in the lungs of rhesus macaques exhibiting clinical symptoms of severe TB disease [52], (iii) whether the gene is significantly up- or down-regulated in the blood of individuals with active TB [54], and (iv) whether the gene is differentially expressed in inbred mouse lungs across variable host genotypes, *Mtb* strains, and infectious doses [53]. To be included in the heatmap, genes were required to encode proteins and to contain a known SNP from the NOD inbred line [55]. Mouse chromosome 2 image generated in the R package karyoploteR. **(B)** CTSZ protein was measured from the lung homogenate of uninfected B6 and the *Tip5*ˢ CC strains CC033 and CC038 (*n* = 3 mice per genotype). Each lane is a separate biological replicate. Vinculin served as the loading control. The assayed proteins are indicated by black arrows. Relative abundance of the **(C)** pro-CTSZ and **(D)** mature CTSZ protein between B6 and the *Tip5*ˢ CC strains, quantified from Fig 1B by normalizing CTSZ levels for each biological replicate to its respective vinculin level. Values plotted as a percentage of the mean CTSZ to vinculin band intensity ratio relative to the average ratio for B6 mice. Hypothesis testing was performed by one-way ANOVA and Dunnett's *post hoc* test on individual ratios between CTSZ and vinculin band intensities by genotype. **(E)** Bacterial burden measured from lung homogenate 4 weeks after aerosol infection with *Mtb* H37Rv (*n* = 3–12 mice per strain; all males except B6 and *Ctsz*⁻/⁻ groups, which included both sexes in equal proportion). Hypothesis testing was performed by one-way ANOVA and Dunnett's *post hoc* test on log₁₀-transformed values. The data underlying this figure can be found in S1 Data sheets 1C, 1D, and 1E.

transcription was significantly repressed [52,54]. In an additional lung transcriptomic study in inbred mice leveraging distinct *Mtb* strains and infectious doses, only 5 gene transcripts within *Tip5*, including *Ctsz*, were differentially regulated across all strains and doses [53]. Currently, there is no established association between human *ZNF831* SNPs and TB outcomes. Conversely, mutations in human *CTSZ* were previously associated with poorer TB outcomes [23,24,26]. From this analysis, *Ctsz* was identified as a lead candidate for further interrogation as a potential genetic cause of *Tip5*-linked TB susceptibility.

## The susceptible NOD variant of *Tip5* and ablation of *Ctsz* both impart TB susceptibility

To evaluate *Ctsz* as a causal factor underlying *Tip5*-linked susceptibility, we measured CTSZ protein from the lungs of uninfected CC strains harboring the susceptible NOD *Tip5* variant (CC033, CC038). Compared to *Mtb*-resistant B6, the

lungs of both CC033 and CC038 exhibited significantly lower baseline levels of CTSZ protein (Fig 1B), both in the pro-form (Fig 1C) and mature active form (Fig 1D). Collectively, these data suggest that the NOD *Tip5* haplotype contains a hypomorphic variant of *Ctsz*, resulting in reduced production of CTSZ protein in *Tip5$^S$* CC strains.

Considering the *Tip5* QTL was first identified in a large-scale in vivo screen, we next assessed whether *Tip5$^S$* CC strains and *Ctsz* null mice (*Ctsz$^{-/-}$*) (S1A Fig) are susceptible to aerosol infection, the natural route of *Mtb* infection. A cohort including B6, CC033, CC038, *Ctsz$^{-/-}$*, and highly susceptible interferon gamma receptor null mice (*Ifngr$^{-/-}$*) [56] was infected via aerosol route with *Mtb* H37Rv. The experiment terminated at 4 weeks postinfection, after the onset of adaptive immunity [57] and matching the initial CC screen endpoint [50]. Relative to B6, all infected strains exhibited significantly higher pulmonary *Mtb* burden (Fig 1E). The CC strains exhibited 10-fold greater lung CFU than B6, surpassing the canonically susceptible *Ifngr$^{-/-}$* mice. *Ctsz$^{-/-}$* mice exhibited a 2-fold increase in lung burden relative to B6. No significant differences were identified in disseminated spleen burden at this time point (S1B Fig). We conclude that *Tip5$^S$* CC strains and *Ctsz$^{-/-}$* mice exhibit reduced pulmonary bacterial control at 4 weeks postinfection.

### *Ctsz* mediates lung CXCL1 levels early during *Mtb* infection

To characterize the impact of *Ctsz* on disease progression, we infected B6 and *Ctsz$^{-/-}$* mice via aerosol, sacrificing cohorts of mice at 2, 3, 4, and 8 weeks postinfection to capture innate and adaptive immune responses. *Ctsz$^{-/-}$* mice exhibited higher lung burden at 2 weeks (4.09 $\log_{10}$ CFU versus 3.41 in B6; $p < 0.05$) and 4 weeks (5.17 $\log_{10}$ CFU versus 4.09 in B6; $p < 0.05$) postinfection (Fig 2A). Similarly, at 3 weeks postinfection, *Ctsz$^{-/-}$* mice exhibited trends toward elevated spleen burden (2.68 $\log_{10}$ CFU versus 2.17 in B6; $p = 0.058$), suggesting earlier dissemination and weaker bacterial containment in the lungs of *Ctsz$^{-/-}$* mice (Fig 2B). However, by 4 weeks postinfection, spleen burden was indifferentiable between *Ctsz$^{-/-}$* and B6.

To profile the impact of *Ctsz* disturbance on the lung inflammatory response throughout the course of infection, we compared cytokine signatures of *Ctsz$^{-/-}$* with B6 at assayed timepoints. At 4 weeks postinfection, *Ctsz$^{-/-}$* mice exhibited higher concentrations of $T_H1$-associated cytokines, like TNF-α ($p = 0.019$) and IL-1β ($p = 0.016$), and lower levels of GM-CSF ($p = 3.8e^{-06}$), IL-6 ($p = 5.9e^{-04}$), LIF ($p = 6.6e^{-07}$), and VEGF ($p = 6.6e^{-07}$) compared to B6 (Fig 2C).

To identify unique features in the inflammatory signature of *Ctsz$^{-/-}$* mice, we performed sparse partial least squares discriminant analysis (sPLS-DA) across measured phenotypes (Fig 2D). Higher lung burden and CXCL1 levels in *Ctsz$^{-/-}$* mice were the strongest features underlying sparse component 1 (Fig 2E). Although component 1 explains 19% of variance in the data compared to 23% variance explained by component 2 (S2A Fig), component 1 better captures the variance attributable to genotype. CXCL1 has previously been identified as a biomarker of active TB disease in genetically diverse mice [51] and in humans [58]. From 2 to 4 weeks postinfection, *Ctsz$^{-/-}$* mice exhibited significantly higher lung CXCL1 levels (Fig 2F), suggesting that *Ctsz* ablation increases disease severity. However, by 8 weeks postinfection, although mean CXCL1 levels in *Ctsz$^{-/-}$* lungs were elevated, the difference was no longer significant. Enhanced production of CXCL1 was consistent throughout infection, suggesting that this effect may occur independent of differences in *Mtb* burden.

To explore the possibility that elevated CXCL1 levels may occur independent of infection in *Ctsz$^{-/-}$* mice, we sacrificed uninfected mice of both sexes. From lung homogenate, we found elevated levels of CXCL1 in *Ctsz$^{-/-}$* compared to B6 (Fig 2C; $p = 0.007$), suggesting that the connection between *Ctsz* and CXCL1 extends beyond the context of infection. Notably, the total CXCL1 levels in uninfected mice were comparable to levels measured at 2 weeks postinfection.

To determine whether *Ctsz* ablation alone is sufficient to confer susceptibility to aerosolized *Mtb* H37Rv, we conducted two longitudinal challenges of B6 and *Ctsz$^{-/-}$* mice in which mice were sacrificed when IACUC-approved humane endpoints were reached. *Ctsz* ablation was associated with a significant reduction in survival time (Fig 2G), which was driven by male mice (S2B and S2C Fig). Thus, disease progression in a host lacking *Ctsz* is characterized by increased lung *Mtb* burden, elevated lung CXCL1 levels indicative of heightened inflammation, and overall mortality risk.

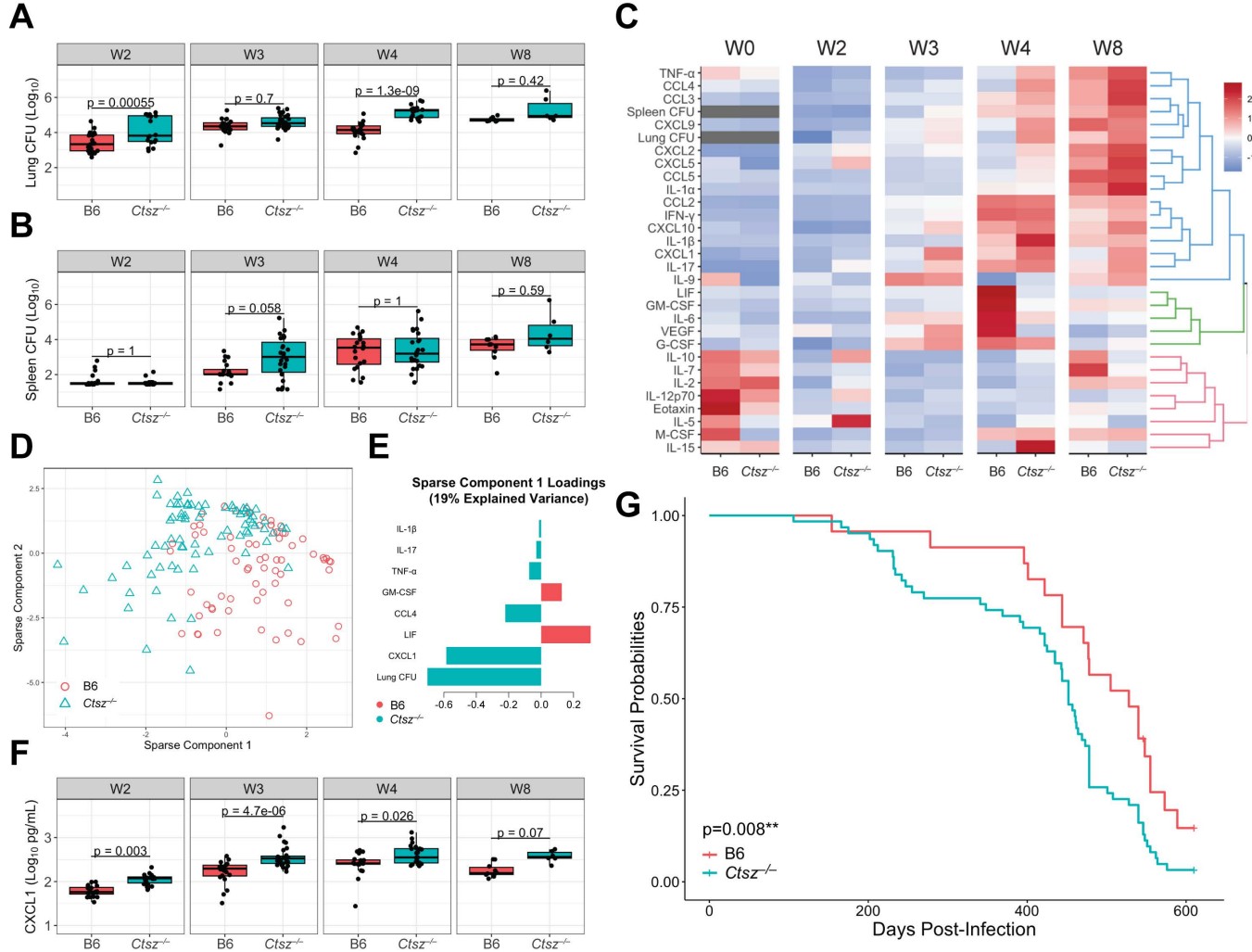

**Fig 2. *Ctsz*<sup>−/−</sup> mice have higher lung burden and earlier spleen dissemination during acute infection followed by greater chronic inflammation and mortality risk.** *Mtb* burden measured from **(A)** lung and **(B)** spleen homogenate by dilution plating. **(C)** Heatmap depicting scaled and centered phenotypes, hierarchically clustered and separated into 3 k clusters. **(D)** Individual mice plotted against the first two sPLS-DA components, which explained the greatest variance in the data after optimization. **(E)** Phenotype loadings contributing to component 1. Component 2 loadings shown in S2A Fig. **(F)** CXCL1 levels measured from lung homogenate by multiplex ELISA. For panels A, B, and F, hypothesis testing was performed by two-way ANOVA and Tukey's *post hoc* test on log$_{10}$-transformed values. For panels A–F, mice were sacrificed at 2, 3, 4, and 8 weeks following aerosolized *Mtb* H37Rv infection. Data are from two independent experiments with $n = 6$–14 mice per genotype, representative of both sexes, at each time point. In panel C, age-matched, uninfected mice ($n = 3$–4 per genotype and sex) were assayed for comparison (designated "W0"). **(G)** Kaplan–Meier survival estimates of aerosol-infected B6 ($n = 23$) and *Ctsz*<sup>−/−</sup> mice ($n = 62$) across two independent experiments. Hypothesis testing was performed using a log-rank test. Equal proportions of both sexes were included. Mice that were not moribund at time of sacrifice were censored for analysis. The data underlying this figure can be found in S1 Data sheets 2ABCDEF_S2A and 2G_S2BC.

## Disturbance of *Ctsz* enhances CXCL1 induction in macrophages

To explore the expression of cathepsin Z across species and mycobacterial infection models, we analyzed two previously published single-cell RNA sequencing (scRNA-Seq) datasets. In zebrafish infected with *Mycobacterium marinum (Mm)*, *ctsz* was most highly expressed in inflammatory macrophages (cluster 9) after 14 days of infection (Fig 3A–3C) [59]. *CTSZ* in cynomolgus macaques was most highly expressed in macrophages 4 weeks after *Mtb* infection compared to other assayed cell types (Fig 3D–3F) [60]. These results agree with literature establishing the presence of CTSZ in

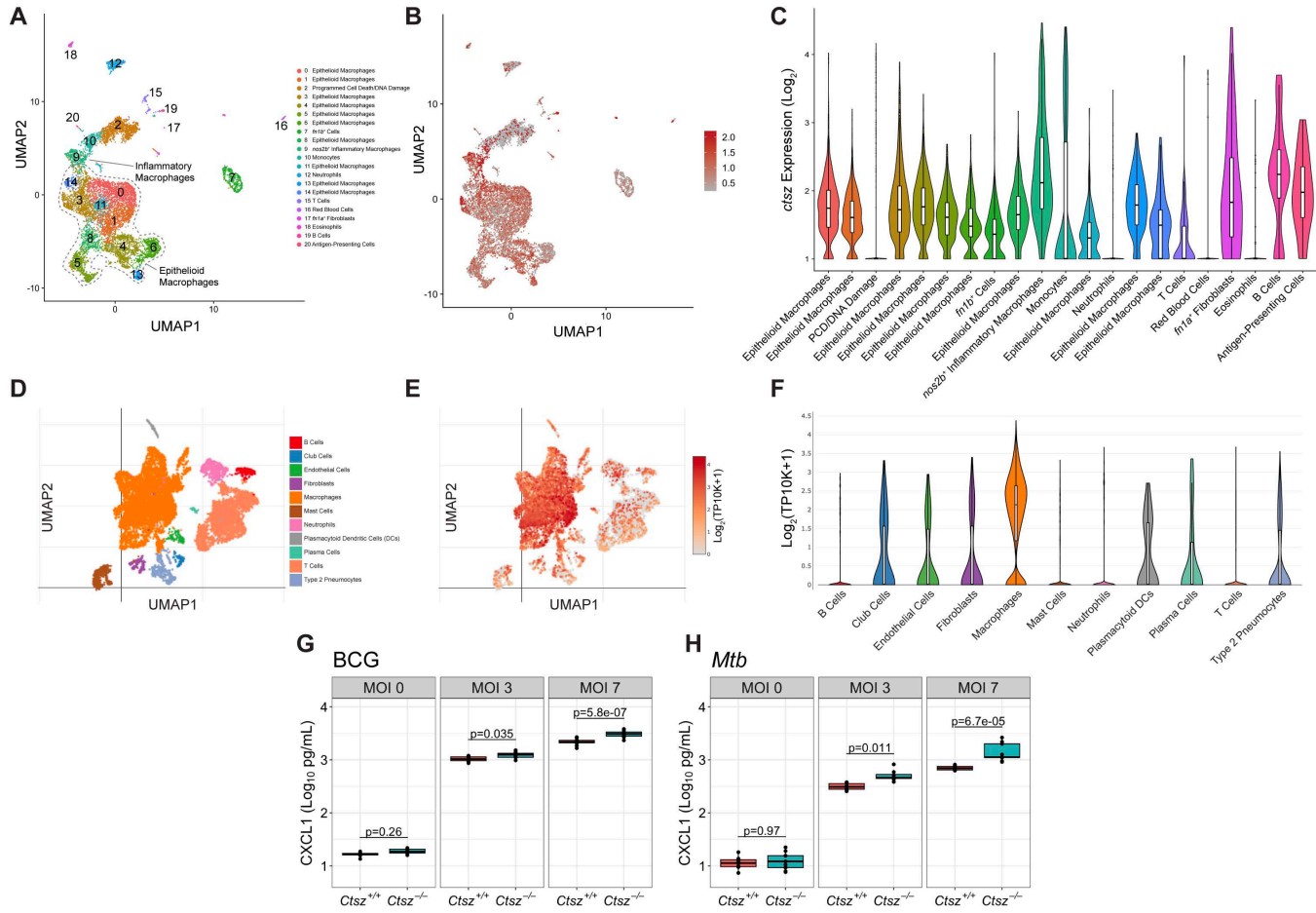

**Fig 3. Cathepsin Z is highly expressed in macrophages across species following mycobacterial infection and mediates levels of CXCL1 in murine macrophages. (A)** UMAP showing scRNA-Seq results from zebrafish granulomas infected with *M. marinum* (*Mm*), generated by reanalysis of previously published data from by Cronan and colleagues [59]. Cells are colored by cluster and assigned in an unsupervised approach from transcriptional signatures. Clusters were annotated by the authors. **(B)** ScRNA-Seq data colored by relative expression of zebrafish *ctsz* in *Mm*-infected granulomas, with highest expression levels observed in cluster 9. **(C)** Violin plots depicting the relative *ctsz* expression by cell cluster, with fill colors and cluster labels aligned with panel **A**. **(D)** UMAP showing scRNA-Seq of granulomas extracted from *Mtb*-infected cynomolgus macaques at 4 weeks postinfection, published by Gideon and colleagues, 2022 [60]. Data accessed at the Broad Institute Single Cell online repository on October 3, 2023 (https://singlecell.broadinstitute.org/single_cell/study/SCP1749/). **(E)** ScRNA-Seq data colored by relative expression of *CTSZ* in *Mtb*-infected macaque granulomas. **(F)** Violin plots depicting the relative *CTSZ* expression by cell type, with fill colors and cell type labels aligned with panel D. CXCL1 levels measured in triplicate at 24 h postinfection from **(G)** BCG-infected and **(H)** *Mtb*-infected BMDMs by ELISA. For panels G and H, BMDMs were differentiated from independent pairs of *Ctsz*^+/+ and *Ctsz*^−/− sibling males for each infection (*N* = 3 infections per pathogen). Dashed threshold denotes the limit of detection for the ELISA. Statistical significance was determined by two-way ANOVA and Tukey's *post hoc* test on batch-corrected, log_10-transformed values. The data underlying this figure can be found in S1 Data sheets 3G and 3H. Data from Cronan and colleagues (2021) and Gideon and colleagues (2022) are available in the NCBI Gene Expression Omnibus (GEO) under accession numbers GSE161712 and GSE200151, respectively.

monocytes and macrophages [27–30] and further highlight that cathepsin Z expression in these cell types following mycobacterial infection is conserved across diverse host species.

As cathepsin Z is consistently expressed in macrophages across several species following mycobacterial infection, we sought to characterize the impact of *Ctsz* ablation on the initial macrophage response to mycobacterial exposure. To test if macrophages contribute to the increased production of CXCL1 during infection in *Ctsz*^−/− mice, we generated BMDMs from *Ctsz*^+/+ and *Ctsz*^−/− sibling pairs. When infected with either nonpathogenic *Mycobacterium bovis* (Bacillus Calmette-Guérin;

BCG) (Fig 3G) or Mtb (Fig 3H), *Ctsz*$^{-/-}$ macrophages produced greater amounts of CXCL1 than *Ctsz*$^{+/+}$ by 24 h postinfection. In both infection models, this effect scaled with increasing multiplicity of infection (MOI). Thus, the elevated CXCL1 we observed in *Ctsz*$^{-/-}$ lungs may be driven by macrophages, especially during the early stages of infection, and appears to be independent of mycobacterial pathogenicity. These results from *Mtb*-infected *Ctsz*$^{-/-}$ mice and BMDMs suggest an interaction between CTSZ and CXCL1 following bacterial exposure.

## Variants in human *CTSZ* are associated with TB severity

To investigate the impact of natural *CTSZ* variation on human TB disease outcomes, we examined whether human *CTSZ* variants are associated with TB disease severity in a household contact study in Uganda ($n = 328$ across two independent cohorts) [61]. Of 81 observed *CTSZ* SNPs, 20 SNPs were associated with differences in Bandim TBScore, a TB severity index (S1 Table; unadjusted $p < 0.05$, linear model with sex, HIV status, and genotypic principal components 1 and 2 as covariates) [62]. After performing a Bonferroni adjustment for multiple comparisons, 4 SNPs and 1 INDEL maintained associations with TB disease severity (Table 1). These variants are in strong linkage disequilibrium (LD) with one another ($R^2 > 0.8$), suggesting that they represent a single haplotype block (Fig 4A). For the most significant SNP (rs113592645), the minor T allele is associated with decreased TB disease severity between those with homozygous major C allele and heterozygotes (Fig 4B, results for other haplotype SNPs included in S3A–3C Fig). To investigate the potential impact of the TB severity SNPs on *CTSZ* expression, we used published RNA-Seq data [63] to compare *CTSZ* transcript levels across *Mtb*- and mock-infected monocytes between genotypes at each *CTSZ* SNP. In the cohort of human-derived monocytes, *CTSZ* was highly expressed at baseline and was downregulated following *Mtb* infection (Fig 4C). Conversely, the rs113592645 minor T allele was associated with increased *CTSZ* expression following *Mtb* infection ($p = 0.0395$; Fig 4D; other haplotype SNP results in S3D–3F Fig). This effect was not observed following mock infection conditions ($p = 0.108$; Fig 4D). Together, these data suggest that these *CTSZ* variants are associated with both TB disease severity and divergent transcription of *CTSZ* following *Mtb* infection.

## CTSZ is produced in macrophages associated with human pulmonary granulomas

The mycobacterial granuloma is an organized structure that can develop within human hosts to contain and restrict *Mtb* infection and is composed of heterogeneous immune cell populations, predominantly macrophages [66]. To investigate whether *CTSZ* expression in macrophages is conserved between mice and humans, we reanalyzed published, spatially resolved scRNA-Seq data from human *Mtb* granulomas [65]. In pulmonary granulomas biopsied from three patients with TB, *CTSZ* was highly upregulated in macrophage cell clusters (Fig 4E–4G). Within patient-derived pulmonary granuloma sections, areas of elevated *CTSZ* expression were found to coincide with regions dominated by macrophages (Fig 4H).

**Table 1. *CTSZ* SNPs significantly associated with TB severity in Ugandan household contact study cohorts, sorted by ascending *p*-value.** Included SNPs were significantly associated with Bandim TBScore after Bonferroni correction for multiple comparisons ($p < 0.05$). Complete collection of 81 SNPs can be found in S1 Table. SNPs are annotated as described in McHenry and colleagues [61]. Allele effects were assessed using a linear mixed effect model in the R package kimma to account for sex, HIV status, and genotypic principal components 1 and 2. Cohorts 1 and 2 are independent cohorts of culture-confirmed adult TB cases. Abbreviations: SNP, single-nucleotide polymorphism; CHR, chromosome; BP, base pair from GRCh38 build; Adj., adjusted; MAF, minor allele frequency.

| SNP | CHR:BP | Effect Allele | Adj. *p*-value | β | MAF in Cohort 1 ($n = 149$) | MAF in Cohort 2 ($n = 179$) |
|---|---|---|---|---|---|---|
| rs113592645 | 20:59001340 | T | 0.0001814 | −1.0036 | 0.18 | 0.061 |
| rs111630627 | 20:59002589 | G | 0.0003077 | −0.9268 | 0.18 | 0.075 |
| rs138964736 | 20:59002671 | ACTTTG | 0.0003077 | −0.9268 | 0.18 | 0.075 |
| rs76687632 | 20:59002905 | G | 0.0003077 | −0.9268 | 0.18 | 0.075 |
| rs8120779 | 20:59001977 | G | 0.0003942 | −0.8671 | 0.18 | 0.095 |

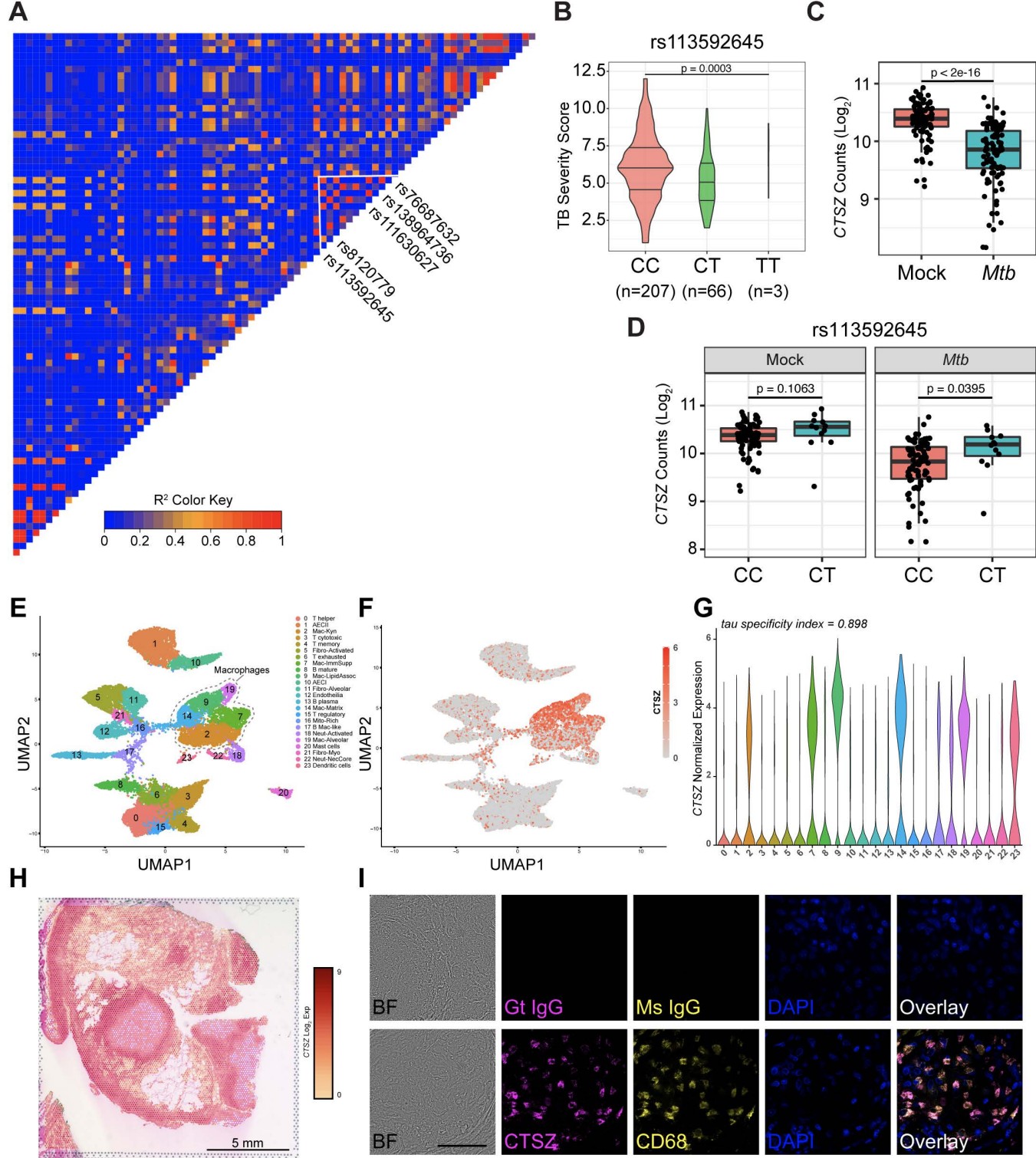

**Fig 4. Human *CTSZ* variants are associated with TB severity, and CTSZ is present at the host-pathogen interface within human pulmonary *Mtb* granulomas. (A)** LD plot of human *CTSZ*, highlighting a haplotype block of 4 identified SNPs and 1 INDEL associated with TB disease severity. **(B)** Comparison of TB severity, measured using Bandim TBScore, by genotype for the lead TB severity SNP, rs113592645. TB severity score by genotype for remaining SNPs can be found in S3A–3C Fig. For panels C and D, *CTSZ* expression was profiled by RNA-Seq in monocytes from 100 Ugandan

individuals. Human-derived monocytes were subjected to 6-hour *Mtb* and mock infection conditions. **(C)** Counts of *CTSZ* transcript (log$_2$ counts per million) collected following mock and *Mtb* infection. **(D)** Counts of *CTSZ* transcript (log$_2$ counts per million) according to genotype for the lead TB severity SNP, rs113592645, following mock and *Mtb* infection. Measurements for homozygous minor allele (TT) were excluded due to low sample size. Counts of *CTSZ* transcript by genotype of remaining SNPs can be found in S3D–3F Fig. **(E)** A manually annotated UMAP generated by unsupervised clustering of data from single-cell mRNA-Seq of human biopsy tissue, containing *Mtb* granulomas from three patients with TB. **(F, G)** Analysis of normalized expression values reveals that *CTSZ* is specifically induced in granuloma macrophages, particularly in lipid-associated macrophages. Values of tau near 1 indicate that *CTSZ* expression is highly specific to some clusters, whereas values near 0 indicate uniform expression across clusters [64]. This expression pattern is part of a broader induction of multiple cathepsins in human *Mtb* granuloma macrophages, shown in S4A Fig. **(H)** The positional distribution of *CTSZ* expression in human *Mtb* granulomas as determined by Visium v2 spatial mRNA-Seq of Eosin-stained biopsy tissue sections from a patient with TB. Similarly processed pulmonary and pleural biopsies from two additional patients with TB can be found in S4B Fig. Panels E–H were generated by reanalysis of data from Pyle and colleagues, 2025 [65]. Cell clusters were annotated by the authors. **(I)** Brightfield (BF) images and immunofluorescent staining of CTSZ and CD68 within a granuloma biopsy from an individual with pulmonary TB. Goat (Gt) and mouse (Ms) IgG isotype control staining is depicted in the top row. DAPI staining indicates cell nuclei. Scale bar is 60 µM in length. Images were captured at 100× magnification. The data underlying this figure can be found in S1 Data sheet 4CD_S3DEF. Data from Pyle and colleagues (2025) are available in the NCBI GEO under accession numbers GSE296399 and GSE296400.

In addition to *CTSZ*, several other cathepsins were also found to be induced in human *Mtb* granuloma macrophages (S4 Fig). To confirm whether elevated *CTSZ* transcription corresponded with elevated CTSZ protein levels in patient tissue samples, we performed immunostaining on granulomas biopsied from the lungs of patients with culture-confirmed TB. We positively identified CTSZ within granuloma-associated CD68⁺ macrophages from *Mtb*-infected lung tissue (Fig 4I). Thus, CTSZ is produced at the site of host-pathogen interaction in humans, suggesting that native functions at this interface could be interrupted should CTSZ production or localization be impeded. Combined with the results from *Ctsz* null mice, these data suggest that balancing cathepsin Z levels is required to regulate lung inflammation and reduce risk of mortality following *Mtb* infection. Collectively, these data establish an association between human *CTSZ* variants and TB disease severity and reveal CTSZ as a granuloma macrophage-associated protein in human lungs.

## Discussion

Over 15 years have passed since the initial discovery that human *CTSZ* is linked with TB disease susceptibility in West and South Africa. However, the relationship between *Mtb* susceptibility and CTSZ had yet to be experimentally determined. We show that genetic interruption of *Ctsz* in mice causes a failure of bacterial restriction and overproduction of CXCL1 during early *Mtb* infection, precipitating an increased risk of mortality. We further show that ablation of *Ctsz* is associated with cell-autonomous overproduction of CXCL1 in macrophages following *Mtb* infection. We report 4 SNPs and 1 INDEL within *CTSZ* significantly associated with TB severity in Ugandan individuals and show elevated *CTSZ* expression in infected monocytes from this cohort. Finally, we find that CTSZ protein is produced within the CD68⁺ macrophages in human granulomas, the pulmonary structure that contains and restricts *Mtb* growth.

CTSZ participates in several known immunological pathways [29,32,33,67]. For example, CTSZ is known to interact with cell surface integrins that mediate immune cell activity, including lymphocyte function-associated antigen-1 (LFA-1) [32,33] and macrophage-1 antigen (Mac-1) [67], which regulates *Mtb* phagocytosis [68] and phagocyte migration. Here, we show that lung CXCL1 levels are consistently elevated in *Ctsz*$^{-/-}$ mice prior to and throughout infection. Moreover, compared to wildtype siblings, *Ctsz*$^{-/-}$ macrophages produce more CXCL1 in response to pathogenic and nonpathogenic mycobacterial infection, suggesting a broad immunological response to bacterial exposure.

CXCL1, a cytokine associated with severe TB disease in mice [51] and in humans [58], is primarily known as a neutrophil chemoattractant [69]. Both *Ctsz* and *Cxcl1* are induced in *Mtb*-infected mice [70]. We are not the first to show a significant increase in CXCL1 levels following pathogenic infection in *Ctsz*$^{-/-}$ mice [71], but to the best of our knowledge, this is the first study to directly link CTSZ and CXCL1 during TB pathogenesis. In a 2022 study, mice with neutrophil-specific ablation of the Mac-1 subunit integrin β$_2$ (CD18) were infected with *Aspergillus fumigatus*, a fungus that is recognized by the immune system through Mac-1 [72]. By 24 h postinfection, Haist and colleagues observed elevated fungal

burden and elevated CXCL1 levels in bronchoalveolar lavage fluid. Similarly, we have observed high *Mtb* burden and CXCL1 in *Ctsz*⁻/⁻ lungs following *Mtb* infection. These results collectively suggest that disturbance of normal CD18 activity, which is known to rely upon *Ctsz* [32,33,67,73,74], may trigger overproduction of CXCL1. Future studies are needed to delineate the implications of this CTSZ-CXCL1 axis with other known roles of CTSZ, including cellular adhesion, migration, and antigen presentation [29].

A deeper understanding of how the functions of CTSZ impact disease severity could prove vital to developing therapeutic strategies for both endogenous and infectious diseases. Men are 1.7-fold more likely to develop active TB than women [75]. Considering the sexually dimorphic mortality risk observed in *Ctsz*⁻/⁻ mice and a previous study reporting that B6 BMDM inflammatory responses are divergent between sexes [76], further study of the sex-specific effects of CTSZ may yield insights into the biology underpinning this imbalance in humans. Beyond TB, CTSZ has been implicated as a mediator of host response during *Helicobacter pylori* infections of patient-derived monocytes and *Salmonella* Typhimurium infections of murine BMDMs [77,78]. Furthermore, mouse and human studies have investigated CTSZ for roles in aging [79,80] and in a number of endogenous conditions, including multiple sclerosis [81], primary biliary cholangitis [82,83], osteoporosis [84], and Alzheimer's [85]. *CTSZ* has also been explored for prognostic value and roles in tumor progression across many cancers [86], including breast [87], colorectal [88], gastric [77], and prostate cancers [89], and hepatocellular carcinoma [90]. Increased *CTSZ* expression was associated with poor patient prognoses in two studies [88,90], with one study proposing *CTSZ* as a putative oncogene [90]. Given the importance of CTSZ across a spectrum of human disease categories, continued study of *CTSZ* may yield insights on the human response to departures from immune homeostasis.

While much remains unknown about the molecular roles of CTSZ during *Mtb* infection, this study is the first, to our knowledge, to identify cathepsin Z as a molecular determinant of TB severity in mice and humans. This study is also the first to report CTSZ localization within granuloma-associated CD68⁺ macrophages in *Mtb*-infected human lungs. Host genetic diversity is a central predictor of TB severity, and consideration of genetic diversity is essential to combat human pathogens as enduring and prolific as *Mtb*.

## Materials and methods

### Ethics statement

All animal studies were conducted in accordance with the guidelines issued in the Guide for the Care and Use of Laboratory Animals of the National Institutes of Health and the Office of Laboratory Animal Welfare. Mouse studies were conducted at Duke University using protocols approved by the Duke Institutional Animal Care and Use Committee (IACUC) (Animal Welfare Assurance #A221-20-11 and #A204-23-10) in a manner designed to minimize pain and suffering in *Mtb*-infected animals. Any animal exhibiting signs of severe disease was immediately euthanized in accordance with IACUC-approved endpoints. Use of patient samples was approved by the Duke University Medical Center Institutional Review Board (IRB) under Protocol #00107795 and the University of Washington IRB under Protocol STUDY00001537. Patient sample processing at Duke University was carried out by Drs. Jadee Neff, Charlie Pyle, and Jason Stout. The human genetic data were obtained from the Kawempe Community Health Study in Uganda, which was approved by the National HIV/AIDS Research Committee of Makerere University (Protocol #014) and the University Hospitals Cleveland Medical Center IRB (Protocol #10-01-25). Final clearance was given by the Uganda National Council for Science and Technology (Ref #658).

### Mice

Male and female C57BL/6J (#000664) and male B6.129S7-*Ifngr1*^tm1Agt^/J (*Ifngr*⁻/⁻; #003288) mice were purchased from The Jackson Laboratory. Male CC033/GeniUncJ (CC033) and CC038/GeniUnc (CC038) mice were purchased from the University of North Carolina (UNC) Systems Genetics Core Facility (SGCF). *Ctsz*⁻/⁻ mice were generously provided by Robin Yates (University of Calgary, Calgary, AB, Canada) and generated as previously described [91]. *Ctsz*⁺/⁺ and *Ctsz*⁻/⁻ mice

were subsequently bred at Duke, using *Ctsz*⁺ᐟ⁻ breeding pairs to enable generation of sex-matched *Ctsz*⁺ᐟ⁺ and *Ctsz*⁻ᐟ⁻ siblings. All mice were housed in a specific pathogen-free facility within standardized living conditions (12-h light/dark, food, and water *ad libitum*). Aerosol-infected mice were matched at 8–12 weeks of age at the time of *Mtb* infection. Mice were individually identified for weighing and wellness assessment throughout infection using Bio Medic Data Systems implantable electronic ID transponders (TP-1000) implanted subcutaneously at the back of the neck prior to infection.

## Genotyping

In-house confirmation of *Ctsz*⁻ᐟ⁻ genotype was performed using forward primer 5′-TTG CTG TTG GCG AGT GCG-3′ and reverse primer 5′-CTT GTC ACC AGA TTC CAG C-3′ to detect wildtype *Ctsz* and forward primer 5′-GCT ACC TGC CCA TTC GAC-3′ and reverse primer 5′-ACA GTA GGA CTG GCC AGC-3′ to detect knockout product. Primer sequences were generously provided by Robin Yates (University of Calgary, Calgary, AB, Canada). DNA was extracted from tissue samples using the DNEasy Blood & Tissue Kit (Qiagen). DNA products were prepared for PCR using Q5 High-Fidelity Master Mix (New England BioLabs) and amplified. Protocol included initial 98 °C (30s), then 34 cycles of denaturation (98 °C, 10s), annealing (68 °C, 30s), and extension (72 °C, 90s), and a final 72 °C (180s), resting at 10 °C ∞ until stopped. Amplified products were separated on a 1% agarose-TAE gel using SYBR Safe stain (Thermo Fisher Scientific) and 1kb Plus DNA Ladder (New England BioLabs). *Ctsz*⁺ᐟ⁺ and *Ctsz*⁻ᐟ⁻ mice were genotyped at the time of weaning from ear and tail tissue biopsies by TransnetYX (Cordova, TN, USA) using proprietary RT-PCR primers designed to detect both *lacZ*, present in the IRES vector disturbing the second exon of *Ctsz* [91], and wildtype *Ctsz*.

## Bacterial strains and culture

All infections were performed with either *Mtb* H37Rv genotype or *M. bovis* BCG (Bacillus Calmette-Guérin) Danish (gift from Sunhee Lee, University of Texas Medical Branch, Galveston, TX, USA), which was transformed with pTEC-15 wasabi fluor and possesses a hygromycin resistance marker for selection [92]. Aerosol infections were performed using an *Mtb* H37Rv strain confirmed to be positive for the cell wall lipid and virulence factor phthiocerol dimycocerosate (PDIM; gift from Kyu Y. Rhee, Weill Cornell Medicine, New York, NY, USA). Bacteria were cultured in Middlebrook 7H9 medium supplemented with oleic acid-albumin-dextrose catalase (OADC), 0.2% glycerol, and 0.05% Tween 80 (or 0.005% tyloxapol for macrophage infections) to log-phase with shaking (200rpm) at 37 °C. Hygromycin (50µg/mL) was added when necessary. Prior to all in vivo infections, cultures were washed and resuspended in phosphate-buffered saline (PBS) containing 0.05% Tween 80 (hereafter PBS-T). Bacterial aggregates were then broken into single cells using a blunt needle before diluting to desired concentration for infection.

## Mouse infections

Mice were infected with ~150−350 *Mtb* CFU via aerosol inhalation (Glas-Col). On the day following each infection, one cage was sacrificed to enumerate lung CFU as an approximation of infectious dose. For all infections, mice were euthanized in accordance with approved IACUC protocols, and lung and spleen were harvested into PBS-T and processed in a FastPrep-24 Homogenizer (MP Biomedicals, 4.0 m/s, 45s, 2–3×). *Mtb* burden was quantified by dilution plating onto Middlebrook 7H10 agar supplemented with OADC, 0.2% glycerol, 50 µg/mL Carbenicillin, 10 µg/mL Amphotericin B, 25 µg/mL Polymyxin B, and 20 µg/mL Trimethoprim. Lung homogenate was centrifuged through a 0.2 µm filter to collect decontaminated filtrate, and cytokines and chemokines were assayed using the pro-inflammatory focused 32-plex assay (Eve Technologies, Calgary, AB, Canada).

## Human tissue immunofluorescent staining

Patient tissue samples containing *Mtb* granulomas were identified at the Duke University School of Medicine. Clinical tissue specimens were obtained from the Duke Pathology Department, and 5 µm paraffin sections for antibody staining were

cut by the Research Histology Laboratory within the BioRepository & Precision Pathology Center (BRPC). Paraffin was dissolved using two xylene washes followed by washes with ethanol of increasing dilution (100% twice, 95% twice, 70% once, and 50% once), three washes with deionized water, and a final wash in PBS. Sample was placed in antigen retrieval buffer (10 mM Tris/1 mM EDTA, pH 9.0) and processed in a pressure cooker for 10 min. Following a cooling step, samples were blocked for an hour in 2.5% normal donkey serum. Samples were incubated overnight at 4 °C with Goat anti-Human/Mouse/Rat Cathepsin X/Z/P Polyclonal Antibody (R&D Systems, AF934, 0.185 mg/mL) and Mouse anti-Human CD68 Monoclonal Antibody (Agilent Dako, M081401-2, 0.185 mg/mL) in 2.5% serum in a humidified chamber. Immunoglobulin G (IgG) isotype controls for background staining (Goat: Biotechne, AB-108-C, 1 mg/mL stock; Mouse:GenScript, A01007, 1 mg/mL stock; Rabbit: Invitrogen, 10500C, 3 mg/mL provided) were also used. Primary antibody was removed with three washes of PBS and two of deionized water. Samples were incubated in Alexa Fluor (AF) conjugated secondary antibody (Thermo Fisher Scientific, 1:500; Donkey anti-Goat IgG AF Plus 647:A32849; Donkey anti-Mouse IgG AF 488:A-21202; Donkey anti-Rabbit IgG AF 555:A-31572) in 2.5% serum for 1–3 h. Following three PBS washes, the samples were mounted for imaging in DAPI Fluoromount-G (Southern Biotech, 0100-20) on glass slides (Fisher Scientific, 22-035813). All antibodies used for staining were centrifuged at 10,000 RCF (4 °C) for 10 min to remove antibody precipitate prior to use.

### Microscopy analysis

Human samples were imaged at 100× on a Zeiss Axio Observer Z1 inverted microscope with an X-Light V2 spinning disk confocal imaging system (Biovision). Images were processed identically within Fiji software (v2.14.0/1.54f) for image clarity.

### Bone marrow-derived macrophage infections

$Ctsz^{+/+}$ and $Ctsz^{-/-}$ sibling pairs were sacrificed in accordance with approved IACUC protocols between 10 and 12 weeks of age. For BCG infections, bone marrow was flushed from hip and leg bones with DMEM (Corning) and cultured for a week at 37 °C in a sterile solution of DMEM with 10% heat-inactivated fetal bovine serum (Corning), 18% 3T3-derived M-CSF, 1× Pen/Strep (Corning), and 25 mM HEPES (gibco). For $Mtb$ infections, bone marrow was flushed from hip and leg bones with sterile DMEM (Corning) and frozen in 10% DMSO in heat-inactivated fetal bovine serum (Corning). Aliquots were later thawed and cultured for a week at 37 °C in a sterile solution of DMEM with 10% heat-inactivated fetal bovine serum (Corning), 30 µg/mL recombinant M-CSF (PeproTech), 1X Pen/Strep (Corning), and 25 mM HEPES (gibco). Differentiated macrophages were then plated at a concentration of $3 \times 10^5$ cells/well in a 24-well plate and cultured at 37 °C overnight in a DMEM solution as above but without Pen/Strep. BMDMs were infected with BCG or transported to BSL-3 biocontainment for infection with $Mtb$ at MOI 3 or 7 or left uninfected. Wells were tested for even infection by CFU plating. At 24 h postinfection, supernatants were collected and filtered using a 0.2 µm filter to remove bacteria. Cytokines and chemokines were assayed from using the high-sensitivity 18-plex discovery assay (Eve Technologies, Calgary, AB, Canada).

### Western blotting

For the comparison of mouse CTSZ between uninfected B6 mice and $Tip5^s$ CC strains (CC033 and CC038), whole lungs were collected from male mice (8 weeks of age) into 1 mL of Trizol reagent. Samples were homogenized with sterile beads at 4.5 m/s for 30 s using the FastPrep-24 Homogenizer (MP Biomedicals). For samples in Trizol, protein was precipitated for 15 min using 9 volumes of 100% methanol at room temperature. The protein precipitate was centrifuged at 3,000 rpm for 5 min, dried for 5 min, and washed in an equal volume of 90% methanol. The protein precipitates were then centrifuged for 1 min at 3,000 rpm, dried for 10 min, resuspended in 1 mL of RIPA buffer and 1× Protease Inhibitor Cocktail, and heated for 5–10 min at 95 °C. Equal volumes of each sample were combined with Laemmli Sample Buffer (BioRad) and

2-Mercaptoethanol (BioRad) and heated at 95 °C for 5 min. SDS-PAGE was performed using BioRad Western Blotting kit along with Precision Plus Protein All Blue Prestained Protein Standards (BioRad). Protein was separated using a 4%–20% Mini-PROTEAN TGX Stain-Free Protein Gel (BioRad) and transferred to a polyvinylidene fluoride (PVDF) membrane using a semi-dry transfer protocol on a Trans-Blot Turbo Transfer System (BioRad). Membrane was blocked using EveryBlot Blocking Buffer (BioRad). Primary staining was performed at 4 °C overnight using Human/Mouse/Rat Cathepsin X/Z/P Antibody (R&D Systems; AF934; 1:2,000 dilution in EveryBlot Blocking Buffer). For B6 and *Tip5$^S$* CC mice, 0.1% Tween 20 was added to the blocking buffer and primary staining also included Vinculin (E1E9V) XP Rabbit mAb (Cell Signaling; #13901; 1:5,000 dilution in EveryBlot Blocking Buffer + 0.1% Tween 20). For *Ctsz$^{+/+}$* and *Ctsz$^{-/-}$* mice, secondary staining was performed at room temperature for 60 min using Donkey anti-Goat 680 (LI-COR; 1:20,000 dilution in Every-Blot Blocking Buffer + 0.1% SDS). Blot was washed in TBS-T between blocking and antibody stains, and fluorescence was measured using a LI-COR Odyssey. Secondary staining for B6 and *Tip5$^S$* CC mice was performed at room temperature for 60 min using HRP-conjugated Rabbit Anti-Goat IgG (Proteintech; SA00001-4; 1:5,000 dilution in EveryBlot Blocking Buffer + 0.1% Tween 20) and HRP-conjugated Goat Anti-Rabbit IgG (Proteintech; SA00001-2; 1:5,000 dilution in Every-Blot Blocking Buffer + 0.1% Tween 20). Blot was washed in PBS-T (0.1% Tween 20). Chemiluminescence was developed using SuperSignal West Pico PLUS Chemiluminescent Substrate (Thermo Fisher Scientific) and imaged using a Chemi-Doc Plus Imaging System (BioRad). Quantification of the blot was performed with ImageLab software (version 6.1).

## Human *CTSZ* analysis

We queried the summary statistics from a published genome-wide association study (GWAS) of TB severity in cases from Kampala, Uganda [61]. Briefly, two independent cohorts of culture-confirmed adult TB cases ($n = 149$, $n = 179$) [93] were included in a GWAS. TB severity was quantified using the Bandim TBscore, which enumerates TB symptoms (e.g., cough, hemoptysis, dyspnea) and clinical signs (e.g., anemia, low body mass index, high body temperature) [62,94]. SNPs within *CTSZ* were identified using a 5kb flanking region around the *CTSZ* start and end positions reported in Ensembl (GRCh38). Pairwise LD for these SNPs was evaluated as the squared inter-variant allele count correlations ($R^2$) using PLINK (version 1.90) in the larger of the two cohorts ($n = 179$). An LD plot was generated from these pairwise LD measures using the R package LDheatmap (version 1.0-5) [95]. The model used to estimate allele effects accounted for sex, HIV status, RNA-Seq batch, and genotypic principal components 1 and 2.

SNP eQTL assessment was performed for the four significant SNPs indicated in Table 1. A linear mixed effect model was developed in the R package kimma [96] to compare baseline, media, and *Mtb*-induced *CTSZ* expression against each SNP genotype. The eQTL model accounted for sex, HIV status, RNA-Seq batch, and genotypic principal components 1 and 2. *CTSZ* expression as $\log_2$ (counts per million) was obtained from RNA-Seq data normalized using voom [97]. RNA-Seq data used for these analyses originated from a previously published dataset of CD14$^+$ monocytes isolated from individuals within the Uganda cohort [63]. Monocytes were subjected to 6-hour media or *Mtb* stimulation and transcriptionally assayed.

## Statistical analysis and data visualization

Hypothesis testing was performed using R statistical software (version 4.3.1). Statistical tests used for hypothesis testing are noted in the figure legends. Shapiro–Wilks tests were used to assess normality in phenotype data prior to parametric hypothesis testing, and $\log_{10}$-transformation was applied for normalization where appropriate. Kaplan–Meier survival curves were calculated using the R package survminer (version 0.5.0). A visualization of mouse chromosome 2 was generated using the R package karyoploteR (version 1.16.0) from the GRCm38/mm10 mouse genome build. Heatmaps in Figs 1A and 2C were generated using the R packages ComplexHeatmap (version 2.21.2) and heatmaply (version 1.5.0), respectively. Optimization and sparse partial least squares discriminant analysis (sPLS-DA) on time course infection cohorts were performed on time point data using the R package mixOmics (version 6.24.0).

## Supporting information

**S1 Fig. Genetic validation and infection of *Ctsz*<sup>−/−</sup> mice.** (**A**) Expression of wildtype and truncated *Ctsz* in tail sections from B6, *Ctsz*<sup>+/−</sup>, and *Ctsz*<sup>−/−</sup> mice. Approximate sizes of wildtype and truncated PCR products are indicated by black arrows. As previously described by Sevenich and colleagues, 2010 [91], exon 2 (containing the active site cysteine critical for the enzymatic activity of *Ctsz*), and a portion of intron 3 in *Ctsz* were deleted by homologous recombination and substituted by a cassette comprising an independent ribosomal entry sequence (IRES). External confirmation of these results was obtained by probing the *lacZ* reporter gene present in the inserted IRES vector. (**B**) Bacterial burden measured by dilution plating from spleen homogenate 4 weeks after aerosol infection with *Mtb* H37Rv (*n* = 3–12 per strain; all males except B6 and *Ctsz*<sup>−/−</sup> groups, which included both sexes in equal proportion). Hypothesis testing was performed by one-way ANOVA and Dunnett's post hoc test on log$_{10}$-transformed values. The data underlying this figure can be found in S1 Data sheet S1B.
(PDF)

**S2 Fig. sPLS-DA and survival analysis comparing *Ctsz*<sup>−/−</sup> and B6 mice.** (**A**) Phenotype loadings contributing to sparse component 2. Mice were sacrificed at 2, 3, 4, and 8 weeks after aerosolized *Mtb* infection. Data are from two experiments with *n* = 6–14 mice per genotype, representative of both sexes, at each time point. Kaplan–Meier survival estimates of aerosol-infected B6 (*n* = 23) and *Ctsz*<sup>−/−</sup> mice (*n* = 62) across two independent experiments, among (**B**) male and (**C**) female mice. Hypothesis testing was performed using a log-rank test. Equal proportions of both sexes were included. The data underlying this figure can be found in S1 Data sheets 2ABCDEF_S2A and 2G_S2BC.
(PDF)

**S3 Fig. Minor alleles of *CTSZ* SNPs within the TB severity haplotype block are associated with lower TB severity score and significantly greater *CTSZ* expression.** Comparison of TB severity, measured using Bandim TBScore, by genotype for (**A**) rs111630627, (**B**) rs8120779, and (**C**) rs76687632 SNPs. Expression of each allele of each SNP was assessed by RNA-Seq at 6 h after mock and *Mtb* infection in human-derived monocytes. *CTSZ* expression by monocytes harboring the minor allele for each SNP was significantly increased following both infection conditions for the (**D**) rs111630627, (**E**) rs8120779, and (**F**) rs76687632 SNPs. eQTL effects were assessed with a linear mixed effect model in kimma to account for sex, age, RNA-Seq batch, genotypic principal components 1 and 2, and kinship. The data underlying this figure can be found in S1 Data sheet 4CD_S3DEF.
(PDF)

**S4 Fig. Cathepsin mRNA is highly expressed in human *Mtb* granuloma macrophages.** (**A**) Heatmap depicting mRNA expression levels of several cathepsins and macrophage markers across unsupervised scRNA-Seq cell clusters. (**B**) The positional distribution of *CTSZ* expression in human *Mtb* granulomas as determined by Visium v2 spatial mRNA-Seq of Eosin-stained biopsy tissue sections from two patients with TB. This figure was generated by re-analysis of previously published data from Pyle and colleagues, 2025 [65]. Cell clusters were annotated by the authors. Data from Pyle and colleagues, 2025 are available in the NCBI GEO under accession numbers GSE296399 and GSE296400.
(PDF)

**S1 File. Fig 1B raw image.**
(TIF)

**S2 File. S1A Fig raw image.**
(PNG)

**S1 Data. Source data for main and supporting figures.**
(XLSX)

**S1 Raw images.** **Annotated raw images for Fig 1B and S1A Fig**.
(PDF)

**S1 Table.** **Complete list of 81 *CTSZ* SNPs present in Ugandan household contact study cohorts and their associations with TB severity.** TB severity was evaluated by Bandim TBScore. Summary statistics for the *CTSZ* variants shown are based on a meta-analysis of two independent cohorts of culture-confirmed adult TB cases (described in McHenry and colleagues, 2023 [61]). Each cohort utilized a linear regression model that controlled for HIV status, sex, and one principal component. Unadjusted *p*-values are reported. Abbreviations: CHR, chromosome; BP, base pair from GRCh38 build; MAF, minor allele frequency.
(XLSX)

## Acknowledgments

The authors acknowledge members of the Smith and Tobin Labs for technical expertise; Martin Ferris, Rachel Lynch, and Ginger Shaw for coordination of experimental cohorts of CC mice through the UNC Systems Genetics Core Facility (SGCF); Christopher Sassetti, Douglas Marchuk, and Craig Lowe for thoughtful critiques and suggestions; and the staff of the Regional Biosafety Laboratory at Duke University for ongoing support of personnel safety in BSL-3 biocontainment. We acknowledge the staff of the Duke University BioRepository and Precision Pathology Center (BRPC) for identifying the human clinical cases and collecting and preparing the paraffin tissue sections. The authors also acknowledge the vital contribution of the patients whose samples provided data for this paper and the contributions made by senior physicians, medical officers, health visitors, laboratory personnel, and data personnel working with the Uganda-CWRU Research Collaboration. This study would not be possible without the generous participation of Ugandan patients and families.

## Author contributions

**Conceptualization:** Rachel K. Meade, Dennis C. Ko, Thomas R. Hawn, David M. Tobin, Clare M. Smith.

**Formal analysis:** Rachel K. Meade, Erika J. Hughes, Joshua Ivie, Penelope H. Benchek.

**Funding acquisition:** Catherine M. Stein, Thomas R. Hawn, David M. Tobin, Clare M. Smith.

**Investigation:** Rachel K. Meade, Oyindamola O. Adefisayo, Marco T. P. Gontijo, Summer J. Harris, Charlie J. Pyle, Kaley M. Wilburn, Alwyn M. V. Ecker, Erika J. Hughes, Paloma D. Garcia, Joshua Ivie, Clare M. Smith.

**Methodology:** Clare M. Smith.

**Project administration:** Clare M. Smith.

**Resources:** Michael L. McHenry, Harriet Mayanja-Kizza, Jadee L. Neff, Jason E. Stout.

**Software:** Rachel K. Meade.

**Supervision:** Catherine M. Stein, Thomas R. Hawn, David M. Tobin, Clare M. Smith.

**Validation:** Oyindamola O. Adefisayo, Marco T. P. Gontijo.

**Visualization:** Charlie J. Pyle, Penelope H. Benchek.

**Writing – original draft:** Rachel K. Meade, Oyindamola O. Adefisayo, Marco T. P. Gontijo, Thomas R. Hawn, David M. Tobin, Clare M. Smith.

**Writing – review & editing:** Rachel K. Meade, Oyindamola O. Adefisayo, Marco T. P. Gontijo, Charlie J. Pyle, Kaley M. Wilburn, Alwyn M. V. Ecker, Erika J. Hughes, Paloma D. Garcia, Joshua Ivie, Penelope H. Benchek, Harriet Mayanja-Kizza, Jadee L. Neff, Dennis C. Ko, Jason E. Stout, Catherine M. Stein, Thomas R. Hawn, David M. Tobin, Clare M. Smith.

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
