## [Editor Report · Decision Letter 0]

31 Mar 2025

Dear Dr Smith,

Thank you for submitting your manuscript entitled "Cathepsin Z is a conserved susceptibility factor underlying tuberculosis severity" for consideration by PLOS Biology.

Your manuscript has now been evaluated by the PLOS Biology editorial staff, as well as by an academic editor with relevant expertise, and I am writing to let you know that we would like to send your submission out for external peer review as Short Report, since there is no molecular mechanism. The Academic Editor also mentioned that it would be interesting to find out why is CtsZ and not other cathepsin. This does not mean that this should be addressed before full submission, but I wanted to pass the comment to you.

However, before we can send your manuscript to reviewers, we need you to complete your submission by providing the metadata that is required for full assessment. To this end, please login to Editorial Manager where you will find the paper in the 'Submissions Needing Revisions' folder on your homepage. Please click 'Revise Submission' from the Action Links and complete all additional questions in the submission questionnaire. Please, when adding the rest of the metadata choose "Short Report".

Once your full submission is complete, your paper will undergo a series of checks in preparation for peer review. After your manuscript has passed the checks it will be sent out for review. To provide the metadata for your submission, please Login to Editorial Manager (https://www.editorialmanager.com/pbiology) within two working days, i.e. by Apr 02 2025 11:59PM.

Kind regards,

Melissa

Melissa Vazquez Hernandez, Ph.D.

Associate Editor

PLOS Biology

---

## [Decision Letter · Decision Letter 1]

21 May 2025

Dear Dr Smith,

Thank you for your patience while your manuscript "Cathepsin Z is a conserved susceptibility factor underlying tuberculosis severity" was peer-reviewed at PLOS Biology. It has now been evaluated by the PLOS Biology editors, an Academic Editor with relevant expertise, and by three independent reviewers.

In light of the reviews, which you will find at the end of this email, we would like to invite you to revise the work to thoroughly address the reviewers' reports. As you will see below, majority of reviewers are positive about the relevance and novelty of the study, yet some concerns have raised during revision. Reviewer 1 raises some concerns regarding the Western Blots and wonders why there are sex-dependent outcomes. Reviewer 2 wonders about the bacterial load in the lung, the mechanistical basis for the CTSZ-CXCL1 axis and the reasons for mortality rate after 200 dpi. Reviewer 3 was more negative, saying that the study lacks mechanistic details even for a SR. While Short Reports do not always require mechanistic insights, we do agree with most of the reviewer concerns and would require some additional experimental revisions to address them, as we consider that this would strengthen the work.

Given the extent of revision needed, we cannot make a decision about publication until we have seen the revised manuscript and your response to the reviewers' comments. Your revised manuscript is likely to be sent for further evaluation by all or a subset of the reviewers.

**IMPORTANT - SUBMITTING YOUR REVISION**

*Re-submission Checklist*

*Published Peer Review*

*PLOS Data Policy*

*Blot and Gel Data Policy*

Sincerely,

Melissa

Melissa Vazquez Hernandez, Ph.D.

Associate Editor

PLOS Biology

REVIEWERS' COMMENTS:

Reviewer #1:

Various association studies and some in vitro work have been suggesting an involvement of cathepsin Z (CTSZ) in the susceptibility for mycobacterial infection causing tuberculosis (TB). Based on their previous work on recombinant inbred mouse strains, which identified a TB susceptibility locus on mouse chromosome 2, the authors of this manuscript now identify and validate CTSZ as important mediator of this susceptibility in vivo. This is further connected to the inflammatory response of macrophages, especially increased CXCL1 upon CTSZ deficiency. Finally the paper provides evidence for variants in human CTSZ associated with TB severity in patients. Overall the manuscripts provides important, novel, and mostly convincing information.

Major issues:

1) CTSZ Western blots (Figure 1B/C; Sup. Figure 1B). Background: Cathepsin Z is produced as a pre-pro-protein. After cleaving of the signal peptide in the ER, Western blots usually detect the inactive Pro-CTSZ and the active mature CTSZ. The mature enzyme is generated by cleaving of a 60 amino acid "activation peptide" from the Pro-CTSZ. This is causing a band shift of 4-5 kDa. Some cell types with high CTSZ activity may predominantly (or only) show the active enzyme.

Questions: The Western blots show multiple bands - what bands correspond to which form of CTSZ? What bands were used for quantification shown in Fig 1C? Supp. Figure 1B (intended to proof CTSZ ko) lacks loading controls and shows somewhat different band pattern to Fig 1B.

Recommendation: For detection of mouse CTSZ in mice use R&D AF 1033; the "all-purpose" AF934 is inferior when used on mouse tissue samples.

2) It is striking that the reduced survival in experimental TB is only evident for male CTSZ ko, but not for females of this genotype. Figure 3E+F, analyze macrophages of male mice for CXCL1 production. Is there a CXCL1-difference in female-derived wt and CTSZ ko macrophages? Whatever the outcome it could serve as a starting point to discuss this important sex-dependent outcome (the discussion of this point is not sufficiently elaborated in the current manuscript).

Minor issues:

1) Ctsz-/- Mice: Please give the appropriate full nomenclature of the mice according to the mouse genome informatics database in the method section (as for the IFNg mice). Further note that mice had been generated on a mixed genetic background and were backcrossed to C57BL/6. This has been first reported in PMID: 25274726.

2) Please consider/discuss: The mice have been generated in HM1 ES cells (129P2/OlaHsd background). The mice were carefully backcrossed; but the original background is likely to be maintained around the CTSZ knock-out locus (for which the genotyping selects). How would this affect the results?

3) Supplement. Figure 1A: Please label the PCR bands (what are the specific PCR products?) Please explain in the figure legend to a general reader, why the PCR products form (or not form) in the WT and KO PCRs.

Reviewer #2:

This study by Maede et al. has investigated the impact of Cathepsin Z loss of function on the susceptibility to M. tuberculosis. Their focus on Ctsz comes from previous QTL mapping on Collaborative Cross mice pointing at a small region of chromosome 2 and from converging evidence from human GWAS linking SNPs in Ctsz and susceptibility to TB. This study assembles experimental data produced by the authors mostly on mice but also on human samples and data from previously published study in humans, macaques and zebrafish. They identify CXCL1 overexpression in monocytes/macrophages as a prominent perturbation in Ctsz KO mice. In a human cohort, they show association between TB severity and Ctsz expression in patient-derived monocytes. Finally, they show that Ctsz is expressed in human granuloma macrophages, at the interface between the host and the pathogen.

The merit of this article is to establish the functional impact of genetic variations (null alleles or SNPs) in the Ctsz gene and the susceptibility to TB, and to link it to CXCL1 expression. While the association between severe TB and Ctsz variants has been reported in several studies, experimental evidence of their role was missing.

The results are well presented (with some improvements suggested below) and the conclusions are supported by the data. The article is well-written and structured, especially in regard to the multiple sources and nature of the data. The findings will be interesting to researchers in the TB field and beyond considering the implication of Ctsz in multiple infectious, immune, neurodegenerative or cancer diseases.

Main comments

Although this is not the main focus of the paper, it is interesting to note (Fig 1D) that lung bacteria load is much higher in CC033 and CC038 than in Ctsz -/- mice, suggesting that they carry other susceptibility genes.

The authors mention a "CTSZ-CXCL1 axis" but do not establish a functional link between the two genes, despite the observation that CXCL1 expression is higher in Ctsz-deficient mice. What is the basis for this "axis"? What could be the mechanisms (direct or indirect?) linking these two genes and the potential downstream consequences? Expanding the discussion on this aspect of the work would point at future directions of research.

Ctsz deficiency results in increased lung bacterial burden and CXCL1 expression in the early phase of infection (W2-W4, Fig 2A, 2F) but does no longer at W8 (at least not significantly). However, it results in increased mortality rate starting 200 days p.i. (Fig 2F), as stated in lines 251-253, and 373-375. Could the authors discuss this observation and suggest mechanisms? Do they have data on bacterial burden at later time points that would support chronically (though modestly) elevated bacterial burden? Or could it be due to unfavorable events in the early phase of infection that determine the final outcome?

Minor comments

Lines 86-88: the sentence mixes two distinct points. First, inbred mice (including CC) are genetically identical within strains, which allows replication, investigating sex differences and disease incidence. Second, common laboratory strains were developed from a small pool of progenitors and therefore sample a small amount of genetic polymorphism, much smaller than that of the human population.

Line 100: We therefore sought to determine WHICH genes found

Line 127: Are cathepsin X or P identical to cathepsin Z (confusing nomenclature) or different members of a family?

Line 131: is rhesus monkey ZNF831 homologous to mouse Zfp831? Please explain.

Lines 213-214: specify that uninfected mice are denoted as W0

Line 242, Fig 2C: the heatmap does not show obvious differences between B6 and Ctsz-/- mice. This data should be added as a plot to Fig 2F.

Mine 279, Fig 3E: add the type of infection as a plot label. log-tranform data as in Fig 2F to better visualize low levels.

Line 311 and Fig4B, C, D: show statistical comparison on the plots. In the text, specify that the effect of the T minor allele is based on the comparison between the CC and CT groups.

Lines 319-321: Together, these data suggest that THESE CTSZ variants are associated with both TB disease severity…

Lines 375-376: Explicitly describe the effect. "We show" is exaggerated by comparison with lines 297-298: "suggest an interaction"

Lines 412-414: Do the authors consider Ctsz just as a "correlate"? From the data, they could argue it is a determinant.

Lines 463-466: by crossing heterozygotes to keep the two alleles in the same colony?

Reviewer #3:

The work of Meade et al is an elegant use of mouse genetics to identify the host determinants of susceptibility and resistance to M. tuberculosis (Mtb). In their prior work they had leveraged the genetic diversity of collaborative cross (CC) mice to identify an Mtb susceptibility locus (Tip5). Here they report that one of the genes, cathepsin z (Ctsz), acts to restrict Mtb replication in mice, and they perform analysis on human genetic data that also finds an association of CTSZ with host defense. This study has several strengths, including its basis in sophisticated and rigorous mouse genetic analysis, the leveraging of multiple public datasets to analyze their candidate gene, and integration of new clinical data. However, there are also some weaknesses in this manuscript. The mechanistic details are somewhat sparse, even for a Short Report manuscript. In addition, while the work does make significant advances, it does not quite generate the level of excitement that would appeal to a broad audience needed for publication in a very high-impact journal like PLOS Biology.

There were multiple strengths to this study including a systematic use of public data to narrow candidates (Fig 1A), carefully controlled animal experiments (Fig 1,2), analysis correlating human SNPs to expression of CATZ. Review concerns are listed below.

Major Issues

* While identifying role for Ctsz in vivo an advance for TB researchers, probably not high-impact enough for a general audience. Impact is also somewhat decreased by the existing (though admittedly not very in-depth) work done on Ctsz in macrophages (PMID: 275726050), this group's prior work identifying Tip5 region (PMID: 35112666) as well as prior human genetics work implicating CTSZ in clinical TB (PMID: 21354459)

* Mechanism is somewhat unclear both for how Ctsz is acting, and what the mutation is that is altering its expression in mice. Are there likely SNPs near Ctsz that might be altering expression? Does CTSZ directly restrict Mtb in the BMDMs they generated, or is it acting indirectly in vivo through cytokine release or other means?

*

A. Minor Issues

* Fig 2C- Many of these changes are likely to be consequence of higher bacterial burden rather than CTSZ directly; unclear exactly what is being plotted on heatmap. Log2FC?

* Fig 2G- no statistics shown

* Fig 3- these are relatively modest changes in cytokine levels

* Fig 2 - looks at mouse CTSZ protein levels but not RNA, figure 4 looks at human CTSZ RNA levels but not protein. Would be helpful to link, perhaps by examining CTSZ RNA in mouse lungs or BMDMs alongside protein.

* Fig 4D - these seem like fairly small changes but hard to tell on Log2 scale. Linear scale probably more appropriate with such small differences.

* Fig 4E probably would be useful to show some non-macrophage marker as a control (TCR? CD20?)

---

## [Editor Report · Decision Letter 2]

6 Aug 2025

Dear Clare,

Thank you for your patience while we considered your revised manuscript "Cathepsin Z is a conserved susceptibility factor underlying tuberculosis severity" for publication as a Short Reports at PLOS Biology. This revised version of your manuscript has been evaluated by the PLOS Biology editors, and by the Academic Editor.

Based on our Academic Editor's assessment of your revision, we are likely to accept this manuscript for publication, provided you satisfactorily address the remaining editorial requests. Please also make sure to address the following data and other policy-related requests.

Please supply the numerical values either in the a supplementary file or as a permanent DOI’d deposition for the following figures:

Figure 1CDE, 2A-G, 3CFGH, 4ABCDG, S1B, S2ABC, S3A-F

b) Please cite the location of the data clearly in all relevant main and supplementary Figure legends, e.g. “The data underlying this Figure can be found in S1 Data” or “The data underlying this Figure can be found in https://doi.org/10.5281/zenodo.XXXXX”

c) Please add a scale bar in the following microscopy pictures in Figures: 4I

d) We require the original, uncropped and minimally adjusted images supporting all blot and gel results reported in the Figures 1B, S2A

e) For figures containing FACS data (Figures 3ABDE, 4EF), please provide the FCS files and a picture showing the successive plots and gates that were applied to the FCS files to generate the figure. We ask that you please deposit this data in the FlowRepository (https://flowrepository.org/) and provide the accession number/URL of the deposition in the Data Availability Statement in the online submission form. If FlowRepository is not available, you can upload the files in our system or to a permanent depository like Zenodo

We expect to receive your revised manuscript within two weeks.

*Published Peer Review History*

*Press*

Sincerely,

Melissa

Melissa Vazquez Hernandez, Ph.D.

Associate Editor

PLOS Biology

---

## [Editor Report · Decision Letter 3]

22 Aug 2025

Dear Clare,

Thank you for the submission of your revised Short Reports "Cathepsin Z is a conserved susceptibility factor underlying tuberculosis severity" for publication in PLOS Biology. On behalf of my colleagues and the Academic Editor, Maximiliano Gutierrez, I am pleased to say that we can in principle accept your manuscript for publication, provided you address any remaining formatting and reporting issues. These will be detailed in an email you should receive within 2-3 business days from our colleagues in the journal operations team; no action is required from you until then. Please note that we will not be able to formally accept your manuscript and schedule it for publication until you have completed any requested changes.

PRESS

Sincerely, 

Melissa

Melissa Vazquez Hernandez, Ph.D., Ph.D.

Associate Editor

PLOS Biology
